# Towards Universal Neural Inference

## Abstract

Building general-purpose models that can leverage information across diverse datasets remains challenging due to varying schemas, inconsistent semantics, and arbitrary feature orderings in real-world structured data. We introduce ASPIRE (**A**rbitrary **S**et-based **P**ermutation-**I**nvariant **R**easoning **E**ngine), a universal neural inference model that performs semantic reasoning and prediction over heterogeneous tabular data. ASPIRE combines two key innovations: (1) a permutation-invariant, set-based Transformer architecture that treats feature-value pairs as unordered sets, and (2) a semantic grounding module that leverages natural language descriptions, dataset metadata, and in-context examples to align features across different datasets. This design enables ASPIRE to process arbitrary collections of feature-value pairs from any dataset and make predictions for any specified target without requiring fixed schemas or feature orderings. Once trained on diverse datasets, ASPIRE generalizes to new inference tasks without additional tuning. Our experiments demonstrate substantial improvements: 24% higher average F1 scores in few-shot classification and 71% lower RMSE in regression tasks compared to existing tabular foundation models. Additionally, ASPIRE naturally supports cost-aware active feature acquisition, strategically selecting informative features under budget constraints for previously unseen datasets. These capabilities position ASPIRE as a significant step toward truly universal, semantics-aware inference over structured data, enabling models to leverage patterns across the vast universe of tabular datasets rather than being limited to isolated, schema-specific learning.

## 1 Introduction

The explosion of available datasets across domains—from healthcare and finance to environmental sciences and retail—presents an unprecedented opportunity for machine learning. Terabytes of structured data now exist across thousands of publicly available datasets (Nguyen et al., 2023; Bache & Lichman, 2013; Schuhmann et al., 2022; 2021), yet current machine learning approaches fail to capitalize on this wealth of information. Most models are designed for individual datasets with fixed schemas, *leaving vast amounts of related data on disconnected islands* that cannot inform each other (van Breugel & van der Schaar, 2024; van Breugel et al., 2024).

This limitation is particularly striking when compared to recent advances in other modalities. Foundation models for images and text have demonstrated remarkable success by training on diverse, large-scale datasets where aggregation is straightforward. However, structured tabular data—which represents the majority of real-world data across domains like healthcare, finance, and scientific research—remains largely untapped by foundation modeling approaches. Current foundational knowledge focuses on just *the tip of the iceberg* of data over a few specific modalities, largely ignoring the vast remainder of general tabular data across diverse domains. The core challenge lies in the heterogeneous nature of tabular data: different datasets have varying schemas, inconsistent feature semantics, and no inherent ordering, making it difficult to build models that can leverage patterns across multiple datasets.

Traditional methods, especially gradient-boosted decision trees such as XGBoost (Chen & Guestrin, 2016), have long dominated structured data modeling due to their efficiency and strong empirical performance on individual datasets. Recently, foundation model approaches have emerged for tabular domains. Some studies finetune large language models on serialized table data (Fang et al., 2024; Hegselmann et al., 2023; Gardner et al., 2024), though LLMs are poorly calibrated and struggle with modeling complex continuous distributions autoregressively (Hopkins et al., 2023; Desai & Durrett,

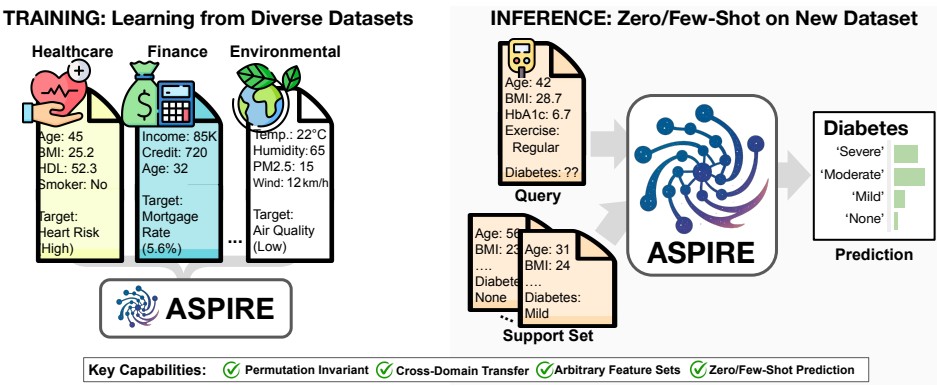

Figure 1: Universal Neural Inference with ASPIRE. During training, ASPIRE learns from diverse datasets across domains with heterogeneous schemas and varying feature sets. The semantic grounding module aligns features using natural language descriptions and metadata. During inference, ASPIRE performs zero-shot or few-shot prediction on new datasets by leveraging learned cross-domain patterns, handling arbitrary feature combinations while maintaining permutation invariance.

2020; Jiang et al., 2023). Others, like TabPFN (Hollmann et al., 2022; 2025) and TabForestPFN (den Breejen et al., 2024), pretrain transformers on synthetic datasets for few-shot prediction. Specialized architectures such as XTab (Zhu et al., 2023), CARTE (Kim et al., 2024), TP-BERTa (Yan et al., 2024) and CM2 (Ye et al., 2024) target cross-table learning across heterogeneous schemas.

Despite these advances, three critical challenges remain. First, schema heterogeneity across datasets—where each dataset differs in feature sets, types, and distributions—impedes model transfer. Second, structured datasets lack inherent ordering, making permutation invariance critical; many existing methods do not guarantee consistent predictions under arbitrary feature arrangements (Arbel et al., 2025). Third, semantic grounding is often overlooked: although column names and metadata vary widely, most models underutilize natural language descriptions that could enable better feature alignment across tables. Recent work emphasizes the importance of conditioning on metadata alongside table data for robust transfer (Klein & Hoffart, 2025).

To address these fundamental challenges, we introduce ASPIRE (Arbitrary Set-based Permutation-Invariant Reasoning Engine), a **Universal Neural Inference** (UNI) framework for heterogeneous structured data, as illustrated in Figure 1. ASPIRE treats tabular inference as a set-based reasoning problem, modeling both features within instances and support examples as unordered sets. This design ensures permutation invariance while enabling flexible reasoning over arbitrary feature combinations. The key innovation lies in ASPIRE's semantic grounding mechanism, which leverages natural language feature descriptions, dataset metadata, and in-context examples to learn meaningful cross-dataset dependencies. By conditioning representations on semantic information rather than feature positions, ASPIRE aligns similar concepts across heterogeneous schemas and generalizes to new datasets without additional training.

This work makes several key contributions to universal tabular reasoning. We formalize the problem of universal neural inference over heterogeneous structured data and propose a principled solution handling *arbitrary feature sets and targets*. Our permutation-invariant architecture guarantees consistent predictions regardless of feature ordering, addressing a critical limitation of existing methods. We demonstrate how natural language descriptions enable robust cross-dataset generalization, allowing models to leverage semantic similarities even with minimal feature overlap.

We evaluate ASPIRE across heterogeneous tabular benchmarks, demonstrating substantial improvements: 24% higher average F1 scores in few-shot classification and 71% lower RMSE in regression tasks compared to leading tabular foundation models. ASPIRE exhibits strong few-shot capabilities without task-specific training and seamlessly extends to active feature acquisition (Ma et al., 2018; Gong et al., 2019; Li & Oliva, 2020; Shim et al., 2018; Li & Oliva, 2021) through its probabilistic framework, optimizing performance under feature-cost constraints. Unlike existing methods requiring retraining for new datasets, ASPIRE supports open-world inference on previously unseen datasets without additional tuning.

In summary, ASPIRE bridges a critical gap in tabular foundation modeling by aligning permutation-invariant architectures with semantic grounding to enable truly universal tabular prediction and reasoning across the vast universe of structured data.

## 2 BACKGROUND: SET MODELING

Set modeling is fundamental to universal tabular inference because real-world structured data naturally exhibits set-like properties: features within instances have no inherent ordering, and the collection of instances in a dataset forms an unordered set. Traditional machine learning approaches that rely on fixed feature orderings fail when applied across heterogeneous datasets where the same semantic concepts may appear in different positions or with different names. By treating tabular data as sets, we can build models that are robust to these variations while maintaining the flexibility needed for universal inference across diverse schemas.

A set is a collection of elements that does not impose any ordering among its members. Models that operate on sets must respect this property. Let $\mathrm{x} = \{x_i\}_{i=1}^n \in \mathcal{X}^n$ denote a set, where $n$ is the cardinality of the set and $\mathcal{X}$ is the domain of each element $x_i$. The following concepts are central to set modeling:

**Definition 1** (Permutation Invariant). A function $f : \mathcal{X}^n \to \mathcal{Y}$ is permutation invariant if, for any permutation $\pi$, we have $f(\pi(\mathrm{x})) = f(\mathrm{x})$.

**Definition 2** (Permutation Equivariant). A function $f : \mathcal{X}^n \to \mathcal{Y}^n$ is permutation equivariant if, for any permutation $\pi$, we have $f(\pi(\mathrm{x})) = \pi(f(\mathrm{x}))$.

Deep neural networks often consist of multiple layers, which can be viewed as compositions of functions. When composing functions over sets, the following properties hold (Zaheer et al., 2017): if $f$ and $g$ are both permutation equivariant functions, then their composition $f \circ g$ is also permutation equivariant. Similarly, if $g$ is permutation equivariant and $f$ is permutation invariant, then the composition $f \circ g$ is permutation invariant.

**Existing Approaches to Set Modeling**  Early methods augment training data with permuted versions to enforce invariance, but this does not guarantee invariance in practice since sequence models exploit positional information (Zaheer et al., 2017). DeepSets (Zaheer et al., 2017) provides a foundational result: any continuous permutation invariant function can be expressed as $f(S) = h\left(\sum_{x \in S} g(x)\right)$, leading to simple two-stage architectures. However, the required latent dimension grows linearly with set size (Wagstaff et al., 2019). Set Transformer (Lee et al., 2019) addresses this limitation by replacing pooling with self-attention mechanisms, which are inherently permutation equivariant. Several extensions have emerged: Hölder-based power means and quasi-arithmetic pooling strategies (Kimura et al., 2024) generalize aggregation functions for increased expressivity, while subset-invariant regularization (Cohen-Karlik et al., 2020) enforces symmetry through learning objectives. Recent theoretical work (Wang et al., 2023) provides refined insights into trade-offs between model capacity and set size. These advances enable richer element interactions while maintaining permutation invariance—principles that directly inspire our ASPIRE architecture.

## 3 PROBLEM FORMULATION

We propose and formalize the problem of **universal inference**: learning a single model that can perform conditional prediction across diverse datasets with heterogeneous schemas, arbitrary feature subsets, and varying targets. Unlike traditional supervised learning that assumes fixed schemas and designated targets, universal inference must handle the fundamental challenges of cross-dataset generalization while maintaining flexibility in both input features and prediction targets.

**Dataset and Instance Representation**  Let $\mathcal{D} = \{\mathcal{D}_k\}_{k=1}^K$ denote a collection of datasets, where each dataset $\mathcal{D}_k = \{e_n^{(k)}\}_{n=1}^{N_k}$ contains $N_k$ independently and identically distributed examples. Each example $e_n^{(k)}$ is represented as a set of $M_k$ feature–value pairs: $e_n^{(k)} = \{(f_m, v_m)\}_{m=1}^{M_k}$, where $f \in \mathcal{F}$ and $\mathcal{F}$ represents the universe of all potential features across datasets. This set-based representation naturally accommodates the heterogeneous nature of real-world datasets where feature sets, types, and semantics vary significantly.

**From Fixed to Arbitrary Conditioning** Traditional supervised learning assumes a fixed schema with a designated target feature, optimizing target prediction given all remaining observed features. However, this rigid formulation limits the model's ability to leverage partial information or adapt to new prediction tasks. We generalize to **universal arbitrary conditioning**, where any subset of features may be observed for a given instance $e_n^{(k)}$, and the task is to predict any subset of unobserved features.

Formally, let $o_n^{(k)} \subseteq \{1, \ldots, M_k\}$ index the observed features and $u_n^{(k)} = \{1, \ldots, M_k\} \setminus o_n^{(k)}$ index the unobserved features. The objective is to maximize the log-likelihood:

$$\log p\big(\{v_m\}_{m \in u_n^{(k)}} \mid \{(f_m, v_m)\}_{m \in o_n^{(k)}}, \{f_m\}_{m \in u_n^{(k)}}\big),$$

which must be permutation-invariant with respect to observed feature–value pairs and permutation-equivariant with respect to unobserved features. Crucially, the observed subset $o_n^{(k)}$ may differ across instances, enabling flexible inference patterns.

**Universal Inference Across Datasets** While prior work has studied arbitrary conditioning within individual datasets (Ivanov et al., 2018; Li et al., 2020; Strauss & Oliva, 2021), we extend this to the cross-dataset setting. Our goal is to learn a universal inference model that performs arbitrary conditional prediction across a distribution of datasets, including previously unseen ones at test time.

To enable cross-dataset generalization, we propose to incorporate semantic context $c_k$ for each dataset $\mathcal{D}_k$, such as natural language descriptions of features and dataset metadata. Additionally, we allow for a few-shot learning by providing an optional support set of labeled examples $S_k = \{e_s^{(k)}\}_{s=1}^{|S_k|}$ from the same dataset. The universal inference objective models the expected conditional distribution over datasets, instances, support sets, and observational patterns:

$$\mathbb{E}_{\mathcal{D}_k \sim \mathcal{D}} \mathbb{E}_{e_n^{(k)}, S_k \sim \mathcal{D}_k} \mathbb{E}_{o_n^{(k)} \sim P(o|e_n^{(k)})} \Big[ \log p\big(\{v_m\}_{m \in u_n^{(k)}} \mid \{(f_m, v_m)\}_{m \in o_n^{(k)}}, \{f_m\}_{m \in u_n^{(k)}}, S_k, c_k\big) \Big], \tag{1}$$

where $P(o \mid e_n^{(k)})$ represents a distribution over observed feature subsets, and $S_k$ is sampled from the same dataset $\mathcal{D}_k$ (excluding the target instance $e_n^{(k)}$). The support set size $|S_k|$ can vary, enabling zero-shot learning when $|S_k| = 0$ and few-shot learning with $|S_k| > 0$.

This formulation captures the essence of universal neural inference: a single model capable of semantically informed, arbitrary conditional inference across diverse datasets. At test time, given any dataset (seen or unseen during training) and an instance $e_n^{(k')}$ with observed features $o_n^{(k')}$, the model must accurately predict the unobserved features $u_n^{(k')}$ by leveraging learned cross-dataset patterns and semantic understanding.

## 4 ASPIRE: ARCHITECTURE AND METHOD

The core challenge of universal inference is handling heterogeneous datasets with varying schemas while ensuring permutation invariance in reasoning over arbitrary feature sets. ASPIRE addresses this through a unified architecture that combines semantic grounding with set-based permutation-invariant processing, as illustrated in Figure C.1.

### 4.1 OVERVIEW

ASPIRE addresses universal inference through a unified architecture that transforms heterogeneous tabular data into shared representations while preserving flexibility for arbitrary conditioning. The architecture builds on a key insight: successful cross-dataset generalization requires both semantic understanding of the dataset and feature meanings and structural dependencies modeling within and across instances.

The transformation proceeds through four interconnected stages. **Semantic Grounding** (§4.2) maps features with similar meanings into a shared semantic space using natural language descriptions and metadata, enabling the model to recognize that "patient_age" and "age_years" represent the same concept across datasets. **Atom Processing** (§4.3) converts feature-value pairs into contextual "atoms" where values are embedded conditionally on their feature semantics, ensuring that "32" is

processed differently for age versus BMI. **Set-Based Instance Representation** (§4.4) treats each instance as an unordered set of atoms, using Set Transformers to capture feature interactions while maintaining permutation equivariance. Finally, **Universal Inference** (§4.5) aggregates information from query instances, optional support sets, and dataset context through hierarchical attention mechanisms, enabling conditional prediction for any target feature while preserving permutation invariance. This design enables ASPIRE to perform *arbitrary conditioning* across datasets with varying schemas, supporting *zero-shot generalization* to unseen domains and *few-shot adaptation* with minimal examples.

## 4.2 Semantic Feature Grounding

To enable cross-dataset generalization, ASPIRE must align features with similar meanings across heterogeneous schemas. A key challenge is that semantically equivalent features often have different names and data types across datasets—for example, BMI might appear as numerical values in one dataset but as discrete categories ("low", "medium", "high") in another, or patient age might be labeled as "Age", "Patient Age", or "Age (years)" across different sources.

In order to achieve semantic awareness, we need a mechanism that can recognize conceptual equivalence despite these surface-level differences in naming conventions and data representations. We propose a semantic grounding framework that maps each feature $f_m$ into a unified representation space by integrating multiple sources of semantic information. Formally, let $\mathcal{M}(f_m)$ denote the set of available metadata for feature $f_m$, which may include natural language descriptions, data type specifications, categorical value sets, and normalization parameters. The semantic feature embedding is constructed as: $\phi(f_m) = \mathcal{F}(\mathcal{M}(f_m))$, where $\mathcal{F}$ is a learnable aggregation function that combines heterogeneous metadata into a unified semantic representation.

In our implementation, we instantiate $\mathcal{F}$ as an additive composition of specialized encoders:

$$\phi(f_m) = E_{\text{desc}} + E_{\text{dtype}} + \mathbb{I}(\text{dtype} = \text{Categorical}) \cdot E_{\text{choices}}, \tag{2}$$

where $E_{\text{desc}}$ encodes natural language descriptions (e.g., using BERT ()), $E_{\text{dtype}}$ provides learnable type embeddings, and $E_{\text{choices}}$ aggregates categorical value representations when applicable. This design ensures that semantically equivalent features (e.g., "Patient Age" and "Age (years)") receive similar embeddings across datasets, enabling effective cross-dataset knowledge transfer.

## 4.3 Feature-Value Atom Processing

Each instance consists of feature-value pairs that we term "atoms"—the fundamental units of information in ASPIRE. A key challenge is achieving contextual awareness: identical values must be embedded differently depending on their feature context. We address this through feature-conditioned value embedding. First, we ensure the value embeddings are adapted to different data types:

$$\nu(v_m|f_m) = \begin{cases} \sum_{c \in f_m.\text{choices}} \mathbb{I}(v_m = c) \cdot \text{BERT}(c) & \text{if categorical} \\ \text{Fourier}(v_m) & \text{if continuous} \\ \mathbf{e}_\emptyset & \text{if missing} \end{cases} \tag{3}$$

where Fourier encoding (Zhou et al., 2025) provides a single-token representation for numerical values, and $\mathbf{e}_\emptyset$ represents a learnable embedding for missing value that is shared across all datasets.

The atom embedding then fuses semantic feature information with contextually encoded values:

$$\psi(f_m, v_m) = \text{AtomMLP}\big(\phi(f_m), \nu(v_m|f_m)\big) \tag{4}$$

The AtomMLP learns to modulate value representations based on feature semantics, ensuring that identical values receive different contextual encodings. For instance, the value "32" will be processed differently when $\phi(f_m)$ indicates "age" versus "BMI", enabling the model to capture feature-specific value interpretations essential for cross-dataset reasoning.

## 4.4 Set-Based Instance Representation

A key innovation in ASPIRE is treating each instance as an unordered set of atoms, $e_n^{(k)} = \{(f_m, v_m)\}_{m=1}^{M_k}$, where $v_m$ might be missing for unobserved features. Rather than compressing this set into a single vector, we employ permutation-equivariant *set-to-set transformations* that maintain the set structure while enabling feature interactions.

Our approach enables *late-fusion* of instance information via permutation-equivariant mappings of the feature-value atoms within an instance. As illustrated in Figure 2, the set-to-set mapping processes atom embeddings to yield multiple output vectors (one per atom) that capture intra-dependencies between co-occurring features. This allows feature representations to become instance-aware without bottlenecking through a single compressed vector.

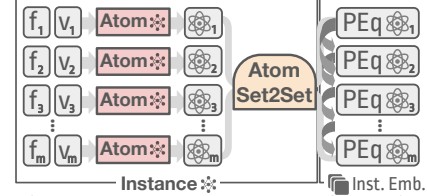

Figure 2: Set-to-set transformation: permutation-equivariant processing of feature-value atoms produces instance-aware representations while maintaining set structure.

Formally, we apply stacked Set Transformer layers (Lee et al., 2019) to the atom embeddings:

$$\rho(e_n^{(k)}) = \text{SetTransformer}\big(\{\psi(f_m, v_m)\}_{m=1}^{M_k}\big). \qquad (5)$$

The Set Transformer architecture is well-suited for this task because both its attention mechanism and feed-forward layers are inherently permutation-equivariant. The final atom embeddings incorporate instance-level contextual information, allowing each feature-value pair to be aware of other features within the same instance.

In few-shot learning scenarios, the model receives a variable number of labeled examples (support set) $S_k = \{e_s^{(k)}\}_{s=1}^{|S_k|}$ from the same dataset. Each support example $e_s^{(k)}$ consists of observed feature-value pairs and is processed through the same instance embedding module, ensuring unified representation across query and support instances:

$$\Lambda(S_k) = \big\{\rho(e_s^{(k)})\big\}_{s=1}^{|S_k|}. \qquad (6)$$

This set-based approach provides two critical advantages: (1) *permutation invariance*—feature ordering does not affect representations, enabling robust handling of heterogeneous schemas, and (2) *flexible conditioning*—any subset of features can be observed or predicted without requiring architectural modifications.

## 4.5 UNIVERSAL INFERENCE ARCHITECTURE

ASPIRE aggregates information from multiple sources through a hierarchical permutation-equivariant framework that processes atoms within instances, then aggregates across instances while maintaining permutation invariance at both levels.

The model processes three types of information:

- **Query instance**: $e_n^{(k)} = \{(f_m, v_m)\}_{m \in o_n^{(k)}} \cup \{(f_m, \emptyset)\}_{m \in u_n^{(k)}}$, where $o_n^{(k)}$ and $u_n^{(k)}$ partition the feature indices into observed and unobserved sets.
- **Support set**: $S_k = \{e_s^{(k)}\}_{s=1}^{|S_k|}$ where each $e_s^{(k)} = \{(f_m, v_m)\}_{m=1}^M$ is fully observed and sampled from the same dataset $\mathcal{D}_k$ as the query instance.
- **Context**: $c_k$ containing dataset description embedded as tokens w/ positional embeddings.

**Hierarchical Permutation-Equivariant Aggregation** ASPIRE employs a two-level architecture where each level maintains permutation-equivariant processing to capture different types of interactions.

**Level 1: Intra-Instance Processing** - Feature-value atoms within each instance are embedded via equation 4 then processed through equation 5 to capture feature interactions while remaining invariant to feature ordering.

**Level 2: Inter-Instance Aggregation** - To enable cross-instance reasoning, we tag each atom representation with learnable type embeddings and aggregate across all sources:

$$\mathcal{T}_{\text{query}} = \{[\rho_m(e_n^{(k)}), \varphi_{\text{query}}]\}_{m \in o_n^{(k)}} \cup \{[\rho_m(e_n^{(k)}), \varphi_{\text{target}}]\}_{m \in u_n^{(k)}} \qquad (7)$$

$$\mathcal{T}_{\text{support}} = \bigcup_{s=1}^{|S_k|} \{[\rho_m(e_s^{(k)}), \varphi_{\text{shot}}]\}_{m=1}^{M_k} \qquad (8)$$

$$\mathcal{T}_{\text{context}} = \{[c_{kt}, \varphi_{\text{context}}]; c_{kt} \in c_k\} \qquad (9)$$

The final aggregation combines all tagged representations through another permutation-equivariant mapping:

$$\mathcal{R} = \text{SetTransformer}(\mathcal{T}_{\text{query}} \cup \mathcal{T}_{\text{support}} \cup \mathcal{T}_{\text{context}}) \qquad (10)$$

---

**Algorithm 1** Training Procedure for Universal Neural Inference Models

---

**Require:** Dataset collection $\mathcal{D} = \{\mathcal{D}_k\}$, masking distribution $P(o|e)$, support size distribution $P(S)$

1: **while** not converged **do**
2:      Sample batch of datasets $\{\mathcal{D}_k\}_{k=1}^{B}$ from $\mathcal{D}$
3:      **for** each dataset $\mathcal{D}_k$ **do**
4:          Sample target instance $e_n^{(k)} \sim \mathcal{D}_k$
5:          Sample observed features $o_n^{(k)} \sim P(o|e_n^{(k)})$              ▷ arbitrary conditioning
6:          Sample support set $S_k \sim \mathcal{D}_k$, where $|S_k| \sim P(S)$      ▷ enables zero/few-shot learning
7:          Sample support examples $\{e_{n'}^{(k)}\}_{n'=1}^{S} \sim \mathcal{D}_k \setminus \{e_n^{(k)}\}$
8:          Compute loss: $\mathcal{L} = -\log p\big(\{v_m\}_{m \in u_n^{(k)}} \mid \{(f_m, v_m)\}_{m \in o_n^{(k)}}, \{f_m\}_{m \in u_n^{(k)}}, S_k, c_k\big)$
9:      **end for**
10:     Update parameters $\theta$
11: **end while**

---

Importantly, this hierarchical design ensures permutation invariance at both levels: atoms can be reordered within instances, and instances can be reordered within the support set, without affecting the final predictions. The natural language context tokens are augmented with positional encodings to preserve the sequential structure. The multi-head attention mechanism enables cross-attention between query, support, and context information for universal tabular reasoning.

**Prediction Heads** From the final unified representation $\mathcal{R}$, we extract embeddings corresponding to target features and apply target-specific prediction heads. Continuous targets are modeled as mixture of Gaussians $\sum_{i=1}^{I} \omega_i \mathcal{N}(\mu_i, \sigma_i)$ where all mixture parameters are predicted from the target embedding. Categorical targets use dot-product attention between the target embedding and pre-computed category embeddings from metadata to produce categorical logits.

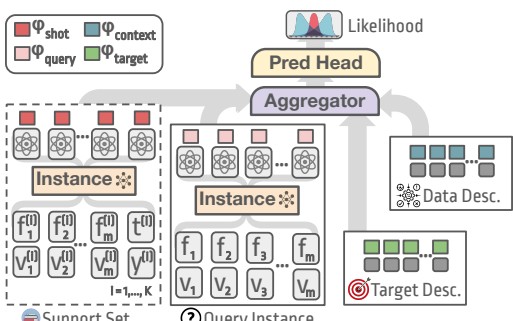

Figure 3: Overview. ASPIRE processes query instances, optional support sets, and dataset context through hierarchical set-based transformations, maintaining permutation invariance at both feature and instance levels for universal tabular inference.

## 4.6 TRAINING PROCEDURE

Optimizing the universal inference objective (equation 1) is challenging due to the intractability of the full expectation over datasets, instances, support sets, and masking patterns. We employ Monte Carlo approximation with a structured sampling strategy that simulates the diverse scenarios ASPIRE will encounter during inference, as detailed in Algorithm 1.

The training procedure incorporates several key design choices that enable universal inference capabilities. The masking distribution $P(o|e_n^{(k)})$ can be uniform (random feature subsets) or reflect realistic missingness patterns to prepare the model for diverse conditioning scenarios. The support size distribution $P(S)$ typically ranges from 0 to 5 examples, enabling the model to handle both zero-shot inference on completely new datasets and few-shot adaptation with minimal examples.

This training strategy exposes ASPIRE to the full spectrum of universal inference challenges: varying dataset schemas, arbitrary feature conditioning patterns, different support set sizes, and diverse prediction targets. The resulting model learns to leverage semantic similarities and cross-dataset patterns while maintaining robustness to the heterogeneous nature of real-world tabular data. Additional implementation details including optimizer configurations, learning schedules, and data preprocessing pipelines are provided in the Appendix. Our code will be open-sourced upon publication.

## 5 RELATED WORKS

**Arbitrary Conditional Models** Arbitrary conditioning is a fundamental problem in density estimation, where the goal is to learn a model capable of predicting any subset of features given any

other subset. Early approaches, such as the Universal Marginalizer (Douglas et al., 2017), use a feed-forward network to predict each unobserved feature independently. Subsequent methods—including VAEAC (Ivanov et al., 2018), ACFlow (Li et al., 2020), and ACE (Strauss & Oliva, 2021)—introduce VAE-, normalizing flow-, and energy-based models, respectively, to better capture dependencies among unobserved features. Neural Conditioner (Belghazi et al., 2019) proposes a GAN-based approach, though it does not provide tractable likelihood estimates. Notably, all these methods are trained on a single dataset and specialize in modeling conditional distributions within that domain. In contrast, our approach, ASPIRE, learns to model arbitrary conditional distributions across a distribution of datasets, enabling broad generalization.

**Tabular Foundation Models**   Recent work has explored building foundation models for tabular data by pretraining on collections of diverse tables. TabPFN (Hollmann et al., 2022; 2025) performs large scale pretraining on synthetic tasks, while CM2 (Yan et al., 2024), XTab (Zhu et al., 2023), and TP-BERTa (Yan et al., 2024) adopt masked modeling objectives to generalize across real-world datasets. Other approaches, such as CARTE (Kim et al., 2024), XTFormer (Chen et al., 2024), and TabLLM (Hegselmann et al., 2023) incorporate graph reasoning, meta-functions, or language modeling to enhance cross-domain transfer. However, most existing methods rely on fixed feature orderings or flatten tables into sequences, which limits their ability to generalize to new schemas. In contrast, ASPIRE maintains both semantic grounding and structural inductive biases such as permutation invariance, enabling robust inference across unseen and heterogeneous tabular domains.

# 6 EXPERIMENTS

We train ASPIRE on $1,400$ upstream tabular datasets from OpenTabs (Ye et al., 2024) with extracted natural language descriptions. We evaluate on 20 downstream tasks from UCI, OpenML, and Kaggle spanning healthcare, finance, and science domains (10-500 features, 1K-1M samples). ASPIRE uses BERT (Devlin et al., 2019) for natural language encoding and token aggregation, and an 8-layer SetTransformer backbone. We report F1 scores for classification and RMSE for regression, averaged over three random seeds. Please look at the appendix D.1 for more details.

**Few-Shot Learning**   We compare ASPIRE against TabPFN, CM2, and LLaMA-3.1-8B-Instruct. LLaMA uses structured prompts containing dataset descriptions, feature descriptions, query instances, and support examples for few-shot prediction. Please see the appendix C.1 for details on the prompt. TabPFN follows the authors' protocol by concatenating support and query instances to obtain posterior predictive distributions, but only supports classification tasks. CM2 is finetuned on the available support examples with early stopping, as it was not designed for in-context few-shot prediction but rather requires adaptation to new datasets as per (Ye et al., 2024).

Table 1 shows F1 scores on classification tasks. ASPIRE consistently outperforms all baselines, improving average F1 scores by 57% over TabPFN and 50% over CM2. The LLM baseline under-

Table 1: Classification performance (F1 scores) across 5-shot, 0-shot and finetuning. Higher is better.

| Dataset | 5-shot | | | | 0-shot | | | | Finetuning | | | |
|---|---|---|---|---|---|---|---|---|---|---|---|---|
| | LLM ↑ | TabPFN ↑ | CM2 ↑ | ASPIRE ↑ | LLM ↑ | TabPFN ↑ | CM2 ↑ | ASPIRE ↑ | MLP ↑ | XGB ↑ | CM2 ↑ | ASPIRE ↑ |
| Diabetes | 0.620 | 0.640 | 0.644 | **0.740** | 0.380 | 0.255 | 0.385 | **0.560** | 0.727 | 0.751 | 0.696 | **0.855** |
| Vehicle | 0.220 | 0.377 | 0.356 | **0.850** | 0.200 | 0.200 | 0.263 | **0.430** | 0.850 | **0.913** | 0.588 | 0.894 |
| Satimage | 0.180 | 0.450 | 0.735 | **0.900** | 0.240 | 0.394 | **0.559** | 0.500 | 0.800 | 0.890 | 0.885 | **0.930** |
| Sick | 0.637 | 0.488 | 0.472 | **0.780** | 0.426 | 0.512 | 0.255 | **0.700** | 0.550 | 0.914 | 0.936 | **0.950** |
| Pc1 | 0.490 | 0.500 | 0.560 | **0.750** | 0.480 | 0.350 | 0.412 | 0.410 | 0.476 | 0.624 | 0.816 | **0.930** |
| Adult | 0.480 | 0.484 | 0.290 | **0.600** | 0.382 | 0.444 | 0.398 | **0.430** | 0.860 | 0.798 | 0.911 | **0.945** |
| Breast | 0.740 | 0.364 | 0.548 | **0.840** | 0.580 | **0.481** | 0.474 | 0.360 | 0.955 | 0.942 | 0.930 | **0.966** |
| Cmc | 0.650 | 0.305 | 0.400 | **0.750** | 0.460 | 0.413 | 0.325 | **0.750** | 0.670 | 0.650 | 0.727 | **0.830** |
| PW | 0.358 | 0.590 | **0.670** | 0.650 | 0.350 | **0.700** | 0.575 | 0.450 | 0.948 | 0.966 | **0.991** | 0.930 |
| Cylinder | 0.368 | 0.518 | 0.390 | **0.710** | 0.295 | **0.600** | 0.400 | 0.390 | 0.532 | 0.734 | 0.814 | **0.883** |
| Mice | 0.330 | 0.476 | **0.590** | 0.500 | 0.300 | 0.200 | 0.250 | **0.480** | 0.995 | 0.990 | 0.980 | **0.999** |
| Car | 0.180 | 0.515 | 0.274 | **0.670** | 0.150 | 0.273 | 0.233 | **0.400** | 0.850 | 0.930 | 0.990 | **0.990** |
| Segment | 0.322 | 0.482 | 0.283 | **0.650** | 0.267 | 0.240 | 0.200 | **0.460** | 0.970 | 0.990 | **0.993** | 0.962 |
| Porto | 0.245 | 0.266 | 0.300 | **0.700** | 0.200 | 0.490 | 0.494 | 0.420 | 0.490 | 0.490 | 0.725 | **0.892** |
| Amazon | 0.463 | 0.438 | 0.700 | **0.730** | 0.460 | 0.520 | 0.411 | **0.530** | 0.870 | 0.483 | **0.970** | 0.876 |
| Average | 0.418 | 0.459 | 0.480 | **0.722** | 0.344 | 0.404 | 0.375 | **0.484** | 0.770 | 0.800 | 0.863 | **0.922** |

Table 2: Regression performance (RMSE) across 5-shot, 0-shot and finetuning. Lower is better.

| Dataset | 5-shot | | | 0-shot | | | Finetuning | | | |
|---|---|---|---|---|---|---|---|---|---|---|
| | LLM ↓ | CM2 ↓ | ASPIRE ↓ | LLM ↓ | CM2 ↓ | ASPIRE ↓ | MLP ↓ | XGB ↓ | CM2 ↓ | ASPIRE ↓ |
| Elevators | 0.432 | 0.973 | **0.256** | 0.733 | 0.940 | **0.524** | 0.360 | 0.340 | 0.292 | **0.097** |
| House Sales | 0.550 | 0.991 | **0.321** | 0.650 | 0.933 | **0.372** | 0.380 | 0.340 | **0.139** | 0.235 |
| Diamonds | 0.844 | 0.920 | **0.211** | 0.862 | 0.927 | **0.337** | 0.344 | 0.203 | 0.970 | **0.020** |
| yprop | 0.561 | 0.866 | **0.312** | 0.717 | 1.110 | **0.464** | 0.987 | 0.982 | 0.977 | **0.228** |
| topo | 0.734 | 0.588 | **0.566** | 0.842 | 0.855 | **0.619** | 0.986 | 0.922 | 0.335 | **0.204** |
| **Average** | 0.624 | 0.867 | **0.333** | 0.760 | 0.953 | **0.463** | 0.611 | 0.557 | 0.543 | **0.157** |

performs due to a lack of tabular specialization. Notably, ASPIRE's zero-shot performance already surpasses CM2's 5-shot results, demonstrating the effectiveness of our universal inference approach and cross-dataset knowledge transfer. Table 2 presents RMSE scores for 5-shot and 0-shot regression. ASPIRE consistently outperforms both the general purpose LLM and tabular specific CM2 baselines, and achieves 72% lower average RMSE than CM2 in 5-shot setting.

**Finetuning**     In addition to few-shot prediction, ASPIRE can be further finetuned on downstream tasks. We evaluate ASPIRE's transferability by finetuning on each dataset's training split and comparing against CM2, MLP, and XGBoost baselines. XGBoost and MLP are dataset-specific baselines, that represent the performance achievable when training dedicated models with full supervision on each individual dataset. Table 1 shows ASPIRE outperforms all baselines on classification tasks, achieving 7% and 15% improvements over CM2 and XGBoost respectively. For regression (Table 2), ASPIRE achieves 71% RMSE reduction compared to CM2. These results demonstrate ASPIRE's strong transferability and effectiveness as a tabular foundation model.

**Ablation Studies**     We systematically evaluate ASPIRE's three key components. **Semantic grounding** enables cross-dataset generalization through natural language descriptions—removing dataset descriptions causes degradation (0.722→0.598 F1). **Permutation invariance** through Set Transformers is crucial, as replacing with independent processing severely degrades performance (0.722→0.381 F1). **Hierarchical aggregation** with learnable type embeddings significantly outperforms alternatives: traditional positional encoding on all aggregation tokens (similar to a traditional LLM) breaks permutation invariance (0.722→0.499 F1) while fixed embeddings prevent adaptation (0.722→0.388 F1). These results demonstrate that all three components are essential for our model. Please see appendix for more details and results on Active Feature Acquisition D.

Table 3: ASPIRE Ablation. F1 and RMSE averaged across datasets.

| Variant | F1 ↑ | RMSE ↓ |
|---|---|---|
| Full ASPIRE | 0.722 | 0.333 |
| **Semantic Grounding** | | |
| w/o Dataset Description | 0.598 | 0.349 |
| **Permutation Invariance** | | |
| Independent (No PEq) | 0.381 | 0.582 |
| **Hierarchical Aggregation** | | |
| Positional Encoding | 0.499 | 0.867 |
| Fixed Embeddings | 0.388 | 0.953 |

# 7    CONCLUSIONS

We introduced ASPIRE, a universal neural inference model that enables cross-dataset generalization for heterogeneous tabular data. By combining permutation-equivariant architectures with semantic feature grounding, ASPIRE treats inference as a set-based reasoning problem conditioned on natural language descriptions and metadata. This design achieves robust generalization across diverse schemas while maintaining permutation invariance—critical requirements that existing tabular foundation models fail to address. ASPIRE demonstrates strong empirical performance, achieving 24% improvement in few-shot F1 scores and 71% reduction in regression RMSE compared to existing methods. The model naturally supports arbitrary conditioning and active feature acquisition, enabling cost-aware inference on previously unseen datasets without retraining. These capabilities represent a significant step toward truly universal tabular reasoning, unlocking the vast potential of structured data across domains rather than constraining models to isolated dataset silos.

ETHICS STATEMENT

This research exclusively uses publicly available datasets and pre-trained models in accordance with their respective licenses and terms of use. No personally identifiable information, sensitive data, or proprietary datasets were collected, generated, or analyzed during this study. All experimental procedures follow standard academic research practices and do not raise ethical concerns regarding data privacy or misuse.

USE OF LLMS

Large language models were used in two capacities in this work: (1) as experimental baselines for comparison with our proposed method, and (2) as writing assistance tools for improving the clarity and presentation of this manuscript. All core technical contributions, experimental design, and analysis were conducted by the authors. The use of LLMs for writing assistance was limited to grammar checking, style improvements, and clarity enhancements, without altering the technical content or conclusions of the research.

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

## A  ADDITIONAL RELATED WORKS

### A.1  SET MODELING

Early methods enforce permutation invariance by augmenting training data with randomly permuted versions of input sets, treating them as sequences while training models to produce consistent outputs across permutations. However, this approach does not guarantee invariance in practice—especially with finite data and limited model capacity—since sequence models inherently exploit positional information (Zaheer et al., 2017).

A foundational work in this area is DeepSets (Zaheer et al., 2017), which proves that any continuous permutation invariant function can be expressed as $f(S) = h\left(\sum_{x \in S} g(x)\right)$, where $g$ maps individual elements and $h$ aggregates the result. This leads to a simple yet expressive two-stage neural architecture. DeepSets also introduces permutation equivariant layers through shared transformations and pooling operations to capture intra-set dependencies. However, later work (Wagstaff et al., 2019) shows that the latent dimension of such architectures must grow at least linearly with the set size to maintain universal approximability, which may limit their practicality.

To model richer interactions between set elements, Set Transformer (Lee et al., 2019) replaces pooling with self-attention mechanisms. Attention layers are inherently permutation equivariant as they compute weighted sums over all elements. By combining these with attention-based pooling, Set Transformer yields permutation-invariant representations while capturing complex intra-set relationships—an approach that lays the foundation to our ASPIRE architecture. Several extensions have emerged to address specific limitations. Hölder-based power means and quasi-arithmetic pooling strategies (Kimura et al., 2024) generalize sum and max pooling for increased expressivity. Other approaches like subset-invariant regularization (Cohen-Karlik et al., 2020) enforce permutation symmetry via learning objectives rather than architectural constraints. Recent work (Wang et al., 2023) provides refined insights into the trade-offs between model width, depth, and set size for maintaining expressivity.

### A.2  ACTIVE FEATURE ACQUISITION

Active Feature Acquisition (AFA) aims to selectively acquire informative features under budget constraints, rather than passively predicting a target from fully observed data. Classical approaches use cost-sensitive classifiers—such as decision trees (Ling et al., 2004), naive Bayes (Chai et al., 2004), and margin-based learners (Nan et al., 2014)—to jointly minimize prediction error and acquisition cost. More recent works frame AFA as a sequential decision-making process and propose various acquisition policies, including greedy information gain (Ma et al., 2018; Gong et al., 2019; Li & Oliva, 2020), tree search (Zubek et al., 2004), imitation learning (He et al., 2012; 2016), and reinforcement learning (Rückstieß et al., 2011; Shim et al., 2018; Zannone et al., 2019; Li & Oliva, 2021; 2024; Li, 2022). However, these approaches are domain-specific and require retraining when applied to new datasets. In contrast, our ASPIRE model enables AFA in an open-world setting—allowing feature acquisition on entirely novel datasets and domains without additional training.

## B  DATA PROCESSING

### B.1  TRAINING DATA

We use datasets from OpenTabs (Ye et al., 2024) for training, validation, and testing. We manually collect dataset descriptions and feature descriptions from UCI ML repository, Kaggle, OpenML, etc. Further, we curate metadata about the dataset by obtaining statistics about the dataset using python functions, for example, collecting potential classes for each target, data type of the feature values etc. Tables which have too few rows and columns, which have unclear and invalid data are dropped. We will opensource this metadata upon publication. We identify task as classification or regression based on target value types, followed by min-max normalization of continuous values. Feature descriptions are embedded using pre-trained transformer models (BERT-base-uncased) to create dense semantic representations that capture feature semantics across diverse domains.

Table B.1: Table of the downstream datasets in our experiments, along with different information

| Dataset Name | R/C | Samples | Numerical | Categorical | Label Classes | Source |
|---|---|---|---|---|---|---|
| Breast | C | 699 | 9 | 0 | 2 | https://archive.ics.uci.edu/dataset/15/breast+cancer+wisconsin+original |
| Bone | C | 1479 | 2 | 7 | 3 | https://archive.ics.uci.edu/dataset/3/connectionist+bench+choice |
| Diabetes | C | 768 | 8 | 0 | 2 | https://openml.org/d/37 |
| Vehicle | C | 846 | 18 | 0 | 4 | https://archive.ics.uci.edu/dataset/149/statlog+vehicle+silhouettes |
| Satimage | C | 6430 | 36 | 0 | 6 | https://archive.ics.uci.edu/dataset/146/statlog+landsat+satellite |
| Sick | C | 3772 | 7 | 22 | 2 | http://archive.ics.uci.edu/dataset/102/thyroid+disease |
| Analcatdata | C | 797 | 0 | 4 | 6 | https://pages.stern.nyu.edu/jsimonof/AnalCatData/Data/ |
| Pcl | C | 1109 | 21 | 0 | 2 | https://openml.org/d/1068 |
| Adult | C | 48842 | 6 | 8 | 2 | https://archive.ics.uci.edu/dataset/2/adult |
| PhishingWebsites | C | 11055 | 0 | 30 | 2 | https://archive.ics.uci.edu/dataset/327/phishing+websites |
| Cylinder-bands | C | 540 | 18 | 21 | 2 | https://archive.ics.uci.edu/dataset/32/cylinder+bands |
| MiceProtein | C | 1080 | 77 | 4 | 8 | https://archive.ics.uci.edu/dataset/342/mice+protein+expression |
| Car | C | 1728 | 0 | 6 | 4 | https://archive.ics.uci.edu/dataset/19/car+evaluation |
| Segment | C | 2310 | 19 | 0 | 7 | http://archive.ics.uci.edu/dataset/50/image+segmentation |
| Porto-seguro | R | 2000 | 26 | 31 | 2 | https://openml.org/d/44787 |
| Amazon | C | 2000 | 0 | 9 | 2 | https://openml.org/d/44712 |
| Elevators | R | 16599 | 18 | 19 | - | https://openml.org/d/216 |
| Yprop | R | 8885 | 251 | 0 | - | https://openml.org/d/416 |
| Topo | R | 8885 | 266 | 267 | - | https://openml.org/d/422 |
| SAT11 | R | 4400 | 115 | 1 | - | https://www.cs.ubc.ca/labs/algorithms/Projects/SATzilla/ |
| Diamonds | R | 53940 | 6 | 3 | - | https://openml.org/d/42225 |
| House_sales | R | 21613 | 20 | 1 | - | https://openml.org/d/42731 |

## B.2 EVALUATION DATA

The following table B.1 detail information about the down stream datasets we use as test datasets in our evaluation.

## C ASPIRE ARCHITECTURE

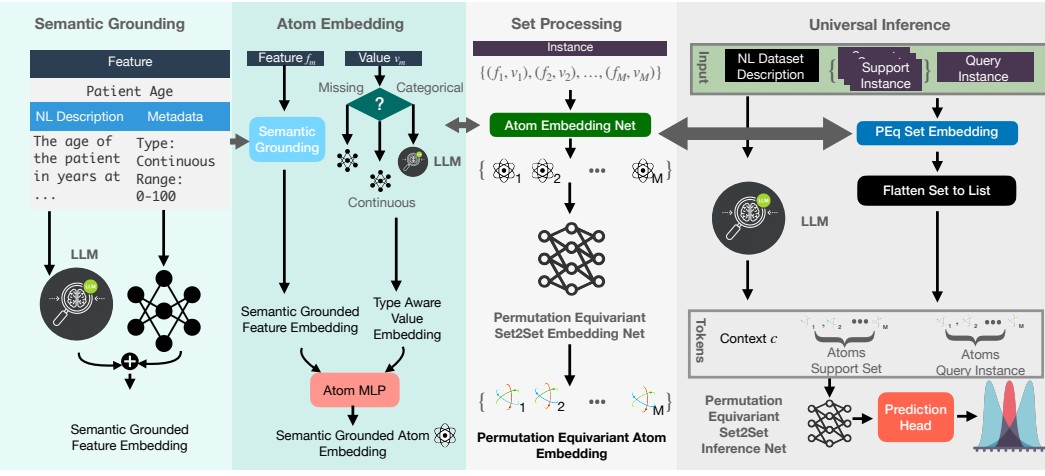

Figure C.1: ASPIRE processing pipeline. The framework transforms heterogeneous inputs from different domains through semantic alignment into a shared understanding space, applies permutation-invariant set processing to capture feature interactions while maintaining order independence, and produces universal predictions that support arbitrary conditioning and cross-domain transfer.

In addition to ASPIRE model descriptions in the paper, the following implementation details are relevant. Our optimization uses AdamW with a learning rate of 1e-4 and cosine annealing with warm restarts (100 warmup steps, weight decay of 0.01-0.04). During training, we apply feature dropout with a 40% masking rate, where selected features have their values replaced with missing value embeddings to improve generalization across diverse tabular domains.

The architecture employs BERT (bert-base-uncased) as the central aggregation mechanism for embeddings from heterogeneous information sources, including dataset descriptions, few-shot examples, and target row representations. Feature encoding uses specialized approaches: real-valued features utilize Fourier embeddings with 256 learnable frequency components spanning 1-256, while categorical

features undergo BERT encoding followed by projection to the model dimension of 768. The Set Transformer components use 4-8 attention heads with 16-32 induced points for permutation-invariant row representations.

The multi-task learning framework combines cross-entropy loss for classification and mixture-of-Gaussians negative log-likelihood for regression, where the regression head employs 10 Gaussian components with learnable weights, means, and log-variances. Training incorporates early stopping with patience of 5-7 epochs. High level pipeline is shown in Fig. C.1

## C.1 BASELINES

Here is the overview of the parameter count of our model and other baselines: ASPIRE Model size - $140,165,123$ parameters CM2 Model size - $53,784,576$. Our model is $2.6\times$ of CM2.**XGBoost**: We implement this using XGBoost package. We set the maximum number of estimators in $50,100,300$. **MLP**: We use 256 dimension hidden layer. Dropout with a rate of 0.1, learning rate 1e-4 and early stopping with patience of 5 epochs. **CM2**: We use the pre-trained model, which have number of transformer layers 3, attnetion heads 8, batch size 256, learning rate for finetuning 3e-4, patience is 5. **TabPFN**: We use batch size 256 and reuse the optimal parameters. For performing few-shot learning, Ye et al. (2024) prescribes $k$-shot learning via finetuning the model with $k$ examples sampled from the train set. We observe that this causes high-variance in the performance, therefore we report the mean of all metrics for CM2 few-shot learning experiments, as shown in Table 1 and Table 2.

The following is the prompt template and example of how we provide Llama-3.1-8B-Instruct with dataset description, feature description and support set for prediction of the query set. We follow a similar format for the regression task.

---

**Classification Prompt**

```
Task:  Classify adult_income data into one of these
categories:  <=50K. or >50K..
Dataset description:  This dataset contains census records
with demographic and employment features, used to predict
whether a person's income exceeds $50K per year.

Features:  age=37, workclass=Private, education=Bachelors,
education_num=13, marital_status=Married-civ-spouse,
occupation=Exec-managerial, relationship=Husband, race=White,
sex=Male, capital_gain=0, capital_loss=0, hours_per_week=60,
native_country=United-States
Label:  >50K.

Features:  age=28, workclass=Private, education=HS-grad,
education_num=9, marital_status=Never-married,
occupation=Handlers-cleaners, relationship=Not-in-family,
race=Black, sex=Male, capital_gain=0, capital_loss=0,
hours_per_week=40, native_country=United-States
Label:  <=50K.

Features:  age=45, workclass=Self-emp-not-inc,
education=Masters, education_num=14,
marital_status=Married-civ-spouse, occupation=Prof-specialty,
relationship=Husband, race=White, sex=Male,
capital_gain=1500, capital_loss=0,n
hours_per_week=50, native_country=United-States
Label:
```

---

# D APPLICATION: ACTIVE FEATURE ACQUISITION

ASPIRE's arbitrary conditioning capability naturally extends to Active Feature Acquisition (AFA), where features are sequentially acquired to minimize cost while maximizing prediction accuracy. Unlike existing methods that require per-dataset training, ASPIRE performs AFA on novel datasets without retraining by leveraging its universal inference framework.

Following prior work (Li & Oliva, 2021), we formulate AFA as a sequential decision-making problem where features are acquired based on estimated mutual information with the target variable. ASPIRE's probabilistic predictions enable principled feature selection under budget constraints, while its cross-dataset knowledge transfer provides robust performance even on previously unseen domains. This demonstrates ASPIRE's flexibility beyond standard prediction tasks, supporting cost-aware inference in open-world settings. Detailed AFA algorithms and experimental results are provided below in this section.

**Active Feature Acquisition** ASPIRE's arbitrary conditioning capability enables active feature acquisition (AFA) without retraining. We evaluate AFA on two classification tasks by sequentially acquiring features and predicting labels at each step. We compare against CM2 and EDDI (Ma et al., 2018), which trains dataset-specific PartialVAE models. Figure D.1 shows results for both finetuned models (left) and 5-shot learning (right). ASPIRE consistently achieves superior performance while requiring fewer acquisitions to reach high accuracy. Notably, ASPIRE's 5-shot performance approaches that of finetuned ones, demonstrating effective cost-aware inference with minimal dataset-specific adaptation.

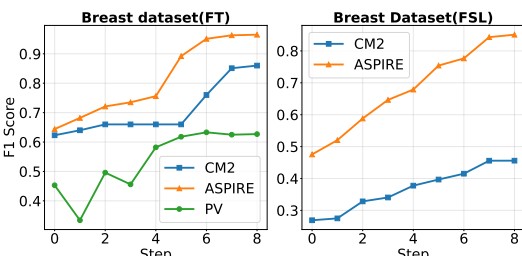

Figure D.1: Active feature acquisition performance (F1 scores). ASPIRE achieves superior accuracy with fewer feature acquisitions. PV: Partial VAE, FT: finetuned, FSL: 5-shot.

## D.1 METHODS

The information reward for acquiring feature $i$ given observed features $\mathbf{x}_o$ is:

$$R(i, \mathbf{x}_o) = \mathbb{E}_{\mathbf{x}_i \sim p(\mathbf{x}_i|\mathbf{x}_o)} D_{KL}[q(\mathbf{z}|\mathbf{x}_i, \mathbf{x}_o)\|q(\mathbf{z}|\mathbf{x}_o)]$$
$$- \mathbb{E}_{\mathbf{x}_\phi, \mathbf{x}_i \sim p(\mathbf{x}_\phi, \mathbf{x}_i|\mathbf{x}_o)} \quad \text{(D.1)}$$
$$D_{KL}[q(\mathbf{z}|\mathbf{x}_\phi, \mathbf{x}_i, \mathbf{x}_o)\|q(\mathbf{z}|\mathbf{x}_\phi, \mathbf{x}_o)]$$

- $\mathbf{x}_o$: Currently observed features
- $\mathbf{x}_i$: Candidate feature $i$ to be acquired
- $\mathbf{x}_\phi$: Target variables (labels)
- $q(\mathbf{z}|\cdot)$: Posterior encoder distribution in VAE
- $p(\mathbf{x}_i|\mathbf{x}_o)$: Conditional distribution of feature $i$ given observed features
- $D_{KL}[\cdot\|\cdot]$: Kullback-Leibler divergence The following algorithm describes our greedy procedure to select the next candidate from the set of unobserved features.

In this algorithm, we use a PartialVAE, but can also be replaced by any other encoder model. For example, hidden representations can be [CLS] token, or target representation token.

## D.2 ADDITIONAL RESULTS

---

**Algorithm 2** Active Feature Acquisition with Partial VAE

---

**Require:** Training dataset $\mathbf{x}_{trn}$, partially observed; Test dataset $\mathbf{x}_{tst}$ without observations; Indices $\phi$
  of target variables.

  1: **TRAINING PHASE:**
  2: Train Partial VAE with $\mathbf{x}_{trn}$
  3: **INFERENCE PHASE (Active Feature Acquisition):**
  4: **for** each test instance **do**
  5:     $\mathbf{x}_O \leftarrow \emptyset$
  6:     **repeat**
  7:         Choose variable $x_i$ from $U \setminus \phi$ to maximize information reward
  8:         $\mathbf{x}_O \leftarrow x_i \cup \mathbf{x}_O$
  9:     **until** stopping criterion reached
 10: **end for**

---

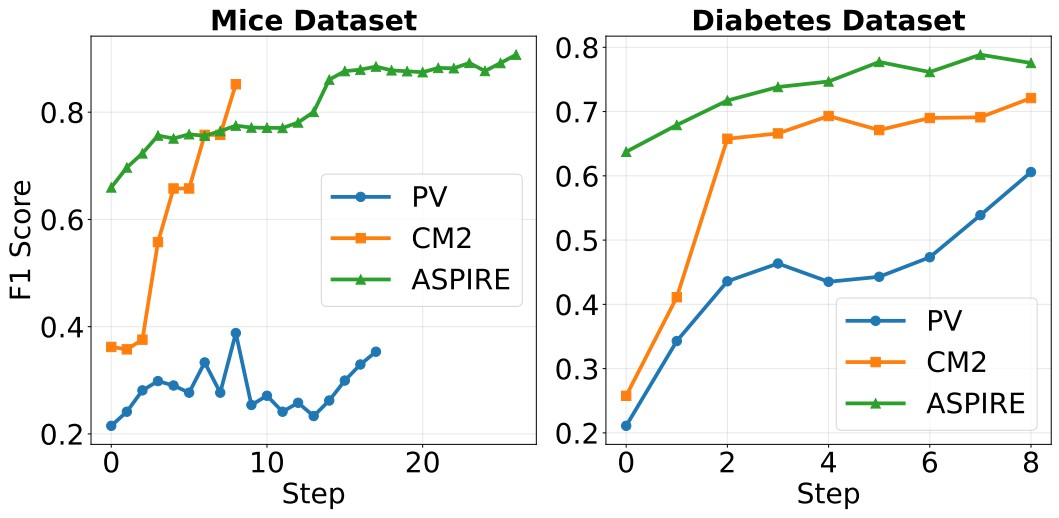

Figure D.2: F1 scores at each feature acquisition step. PV indicates Partial VAE in EDDI. All these models are finetuned on the train split of these datasets.

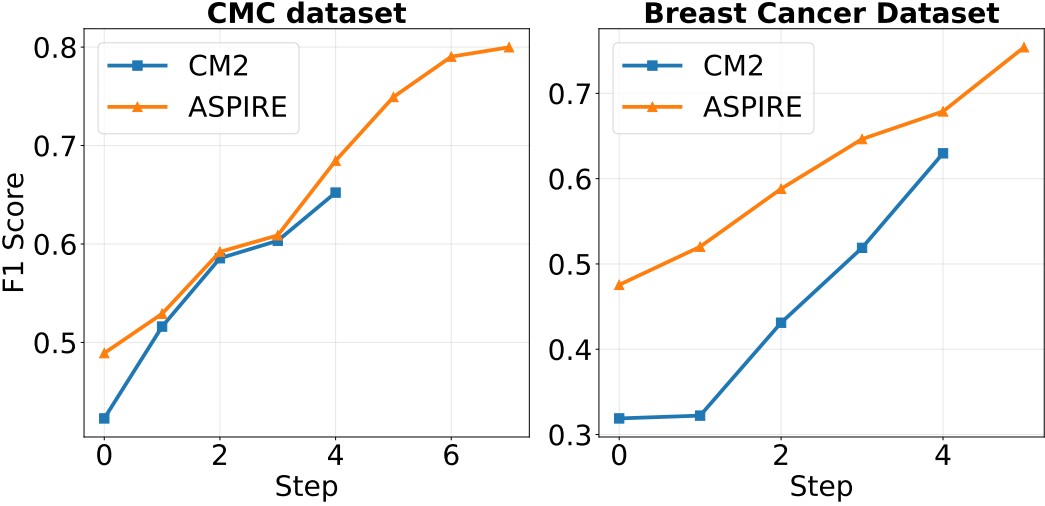

Figure D.3: F1 scores at each feature acquisition step. PV indicates Partial VAE in EDDI. These are the few-shot learning models.

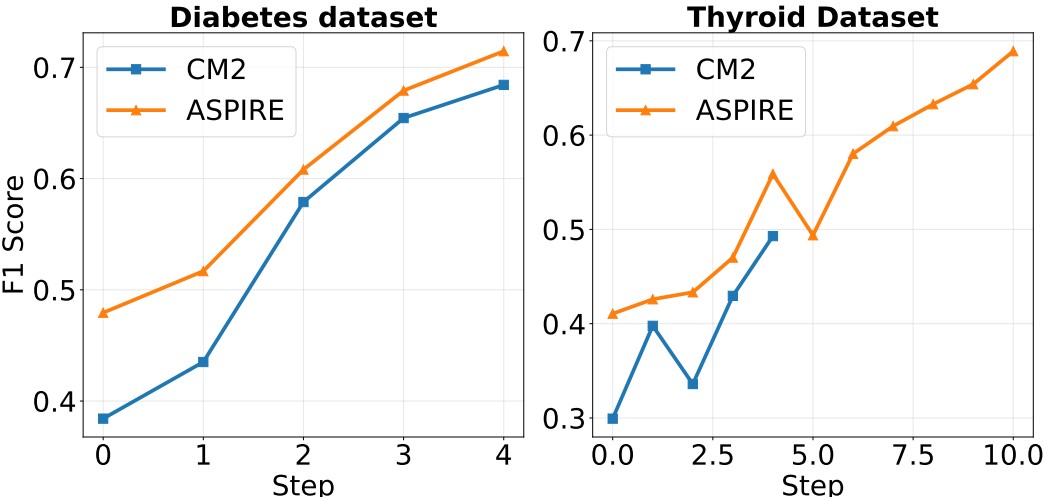

Figure D.4: F1 scores at each feature acquisition step. These are the few-shot learning models.

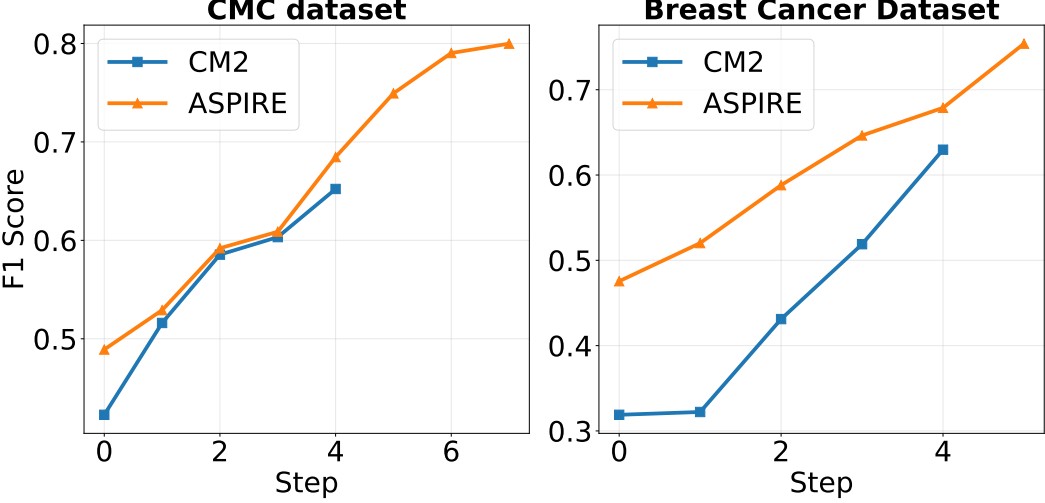

Figure D.5: F1 scores at each feature acquisition step. These are the few-shot learning models

