# OpenReview forum: "Towards Universal Neural Inference"
_ICLR.cc/2026/Conference — ICLR 2026 Conference Withdrawn Submission_

### Official Review · Reviewer_5bLR · 2025-10-26

**Soundness:** 2
**Presentation:** 3
**Contribution:** 1
**Rating:** 2
**Confidence:** 5

**Summary:**

This paper introduces ASPIRE (Arbitrary Set-based Permutation-Invariant Reasoning Engine), which aims to work with heterogeneous tabular data. It tries to solve a few problems: 1. generalize across feature names, meanings 2. work on unseen datasets either with zero-shot or few-shot prediction 3. remain permutation invariant. The core method is a set of different semantic encodings of categorical and continuous features and the usage of set transformer. It conducts simple experiments to show ASPIRE performed better than other models on a number of evaluation datasets.

**Strengths:**

1. The introduction of set modeling: The authors believe the set modeling is the right framework and proposed the framework that handles this modeling approach. However, please see the first point of weakness section as well.
2. The author conducted experiments and compared against LLM, TabPFN and CM2 baselines under few-shot setting which were not done in the prior works.
3. The introduction of fourier transform and set transform together in this setting is novel

**Weaknesses:**

### Originality and Significance
1. The significance of set modeling is not fully justified.
* What is the main purpose of using set modeling and in what specific applications is non-set modeling undesirable?
* Isn't XGBoost or tree-based model already modeling a set?

### Quality and Clarity
1. Missing references: TabICL, TabDPT should be compared to or at least cited.
2. [line 203]: Figure C1 does not appear to be labeled
3. How the fourier transform is done exactly (with or without learned frequency) and how the set transform works should be included in the background.
4. Missing confidence intervals in Table 1 and 2.
5. TabPFN is in 5 shot and 0 shot but not in finetune section. Similarly, XGB is in fine-tune section but not in 5-shot. (Can we still train XGB on just 5 samples?)

**Questions:**

1. Section 4.3: Is the fourier transform parameter learned or fixed? This details needs to be specified.
2. How is TabPFN used for 0-shot? (TabPFN requires at least 1 context sample since the query token does not attend to itself.)
3. Why are F1 score and RMSE chosen for Table 1 and 2? Do we get similar result with AUC and R2?

---

### Official Review · Reviewer_sxLQ · 2025-10-28

**Soundness:** 2
**Presentation:** 2
**Contribution:** 2
**Rating:** 2
**Confidence:** 4

**Summary:**

This paper focuses on the task of constructing foundation model on tabular data. Specifically, the proposed method aggregate semantic grounding, atom processing, set-based instance representation and universal inference to train a foundation model, which then could generalize to different unseen datasets without fine-tuning.

**Strengths:**

S1: The studied problem is important.

S2: The paper structure is clear and easy to follow.

**Weaknesses:**

W1: Towards the motivation. First, the three mentioned challenges seem to be mentioned and solved by previous studies when constructing foundation model. At least, they mentioned these challenges. For example, GTL[1] indicated the first challenge of schema heterogeneity. TP-BERTa[2] indicated the second challenge of feature permutation invariant. GTL[1] also consider the third challenge of semantic grounding. Thus these are the general challenges when constructing tabular foundation model. What unique challenges does this paper solve, which are not solved by previous papers?

W2: Towards the technical novelty. It seems that the proposed method is an alternative solution for these existing challenges compared to previous studies. The proposed method has four main components, i.e., semantic grounding, atom processing, set-based instance representation and universal inference. In detail, many papers, e.g., GTL[1] and TP-BERTa[2] have considered the semantic grounding. TP-BERTa[2] have proposed the similar idea to atom processing, that converts feature-value pair to one unit. TP-BERTa[2] and TabPFNv2[3] also use permutation invariant prediction, which are similar to set-based instance representation.  Finally, the core idea of universal inference detailed in Eq.1, is similar to the core idea of GTL[1]. The authors need to justify what additional improvements or unique advantages they achieved compared to the existing methods.

W3: Towards the technical details. Some details need to be further explained. For example, (1) in what case the output dimension would be the same as input dimension, according to Definition 2? (2) In Eq.2, why E_dtype is learnable? it seems that it only represents numerical or categorical type. If it is learnable, it will have multilple (continues) status rather than binary status. (3) For one specific feature, how are E_dtype and E_choices initialized?

W4: Lack of comparison between sota foudation models. In experiment, is the used TabPFN the first version or the second version? The recent second version of TabPFN is ready for regression. And recently, there are many foundation models following TabPFN, like TabICL[4]. In addition, TP-BERTa[2] and XTab[5], which are mentioned in related work, do not be considered. And maybe also compare with GTL[1].

W5: Lack of comparison between sota few shot learning. At least TabLLM, mentioned in the related work, should be compared, though it only supports classification.

[1] From Supervised to Generative: A Novel Paradigm for Tabular Deep Learning with Large Language Models. KDD'24.

[2] MAKING PRE-TRAINED LANGUAGE MODELS GREAT ON TABULAR PREDICTION. ICLR 2024.

[3] Accurate predictions on small data with a tabular foundation model. Nature 2025.

[4] TabICL: A Tabular Foundation Model for In-Context Learning on Large Data. ICML 2025.

[5] XTab: Cross-table Pretraining for Tabular Transformers. ICML 2023.

**Questions:**

Please see weaknesses above. Need authors to solve all of these problems. As for W4 and W5, the authors should at least give compelling reasons why these baselines are not considered.

---

### Official Review · Reviewer_EdpC · 2025-11-01

**Soundness:** 2
**Presentation:** 2
**Contribution:** 2
**Rating:** 2
**Confidence:** 3

**Summary:**

The paper proposes ASPIRE, a foundational model for classification, regression and imputation for tabular data. The paper pre-trains the model on real data and in particular takes column names, column types and dataset description into account. The authors show that ASPIRE outperforms TabPFN on few-shot prediction on 15 standard benchmark datasets.

**Strengths:**

- The paper adds to the recent work on tabular foundation models, which is a critically important and emerging field.
- The paper in particularly is able to make use of column metadata and dataset metadata, features that are rare in foundational models for tabular data, i.e. they are not present in TabPFN, TabICL and Limix, though they are present to some degree in TabDBT and CARTE.

**Weaknesses:**

- The paper does not compare with any of the recent models that address the same problem, in particular CARTE and TabDPT. Both of these models are able to incorporate column-level meta-data and have been evaluated in the few-shot setting. The only state-of-the-art model evaluated in this paper is TabPFN (assuming this is TabPFN V2, though please clarify if this is the case).

- The presentation of the paper is extremely confusing, in that it emphasizes the set transformer aspect, and the ability to work with varying schemas. Both of these properties are common, and maybe the defining properties of table foundational models, including TabPFN, TabPFNV2, TabICL, Limix, TabDPT, CARTE, and others. This is assuming the goal is to transfer knowledge between datasets with different schemas (which all of these models do), not having varying schemas within the same dataset (which these models do not). If the latter is what is meant in this paper, this should be clarified more, and should be motivated. All of these methods are set transformers with respect to the samples, and some of them are also set-transformers with respect to the features (not TabPFNV1, but TabPFNV2 is, as well as Limix and TabICL, which all have learned embeddings to distinguish columns, so they are not invariant to column reordering. CARTE is invariant IIRC). GAMFormer is completely equivariant with respect to features and samples.

- The masking scheme and universal filling in described this paper is novel relative to TabPFN and TabICL, but very similar to the one used in Limix, and a comparison should be made there.

- CARTE uses a column name embedding somewhat similar to the one described in this paper and a more clear comparison should be made.

- The selection of benchmark datasets is somewhat unclear. Not using a standard benchmark like TabArena, Tabzilla, Talent or OpenML CC-18 makes comparison to other works harder and opens the possibility of dataset selection.

### Minor notes
-  Line 203: the reference to the figure seems broken and it's unclear what figure is references.

- Definition 1: the definition of permutation is unclear. Usually a permutation is a function \pi: {1, ... , n} -> {1, ..., n}. A common form is also writing a permutation matrix P_{\pi}, but in neither case would \pi(x) be defined. I think a definition of \pi(x)_i = x_{\pi(i)} should be added (I assume this is what is meant here, i.e. the coordinates are permuted).

**Questions:**

- Did you use TabPFN V2 or V1 for the comparison?

- Is there a reason not to compare to TabDPT and CARTE that solve the semantic few-shot task as well?

- What is the difference in the masking scheme used in ASPIRE compared to LimiX?

- By "fixed schema" did you mean "fixed schema" across datasets or within a dataset? How is the generalization of ASPIRE different than the one in the other foundational models?

---

### Official Review · Reviewer_khBS · 2025-11-02

**Soundness:** 3
**Presentation:** 3
**Contribution:** 3
**Rating:** 4
**Confidence:** 3

**Summary:**

This paper focues on the tabular data learning and proposes to 1) use set-based transformer to model the arbitrary orderings in tabular data; and 2) leverage meta-information across datasets by a semantic grounding module. The proposed ASPIRE first maps similar features into a shared semantic space, and then develop the "atoms" for each (feature, value) pairs. The set-based transformer is then used to capture feature interactions while maintaining permutation equivariance. Extensive reuslts on tabular data classification and regression tasks show the improvements of the proposed model.

**Strengths:**

1),  The idea that views the tabular data as "atoms" and apply the set transformer to extract the tabular feature is novelty in this community and the results on tabular data show the effectiveness of this idea.

2), The developed two-level aggregation is interesting, and this strategy is helpful to capture differenct information from various levels.

3), The proposed model develops different value embedding methods for categorical and continuous data types, which help to extract the feature more correctly.

**Weaknesses:**

1), One of the main concerns comes from the complex preprocess of semantic feature grounding and feature-value atom processing, which may limit the application of the proposed model in practice.

2), Additional results of TabLLM [1] and FeatLLM [2] need to be inclound to analysis the performance of the proposed model.

3), The choice of datasets is small and non-standard; using a large established benchmark in the literature would lend more credibility to the findings [https://arxiv.org/abs/2305.02997].


[1] Tabllm: Few-shot classification of tabular data with large language models.

[2] Large language models can automatically engineer features for few-shot tabular learning.

**Questions:**

Please see the Weakness section.

---

### Note · Authors · 2025-12-01

I have read and agree with the venue's withdrawal policy on behalf of myself and my co-authors.